# LIN28B and Let-7 in Diffuse Midline Glioma: A Review

**DOI:** 10.3390/cancers15123241

**Published:** 2023-06-19

**Authors:** Truman Knowles, Tina Huang, Jin Qi, Shejuan An, Noah Burket, Scott Cooper, Javad Nazarian, Amanda M. Saratsis

**Affiliations:** 1W.M. Keck Science Department, Scripps, Pitzer, and Claremont McKenna Colleges, Claremont, CA 91711, USA; tknowles23@students.claremontmckenna.edu; 2Department of Neurosurgery, Northwestern University Feinberg School of Medicine, Chicago, IL 60611, USA; tinahuangyt@gmail.com (T.H.); qijin2008@gmail.com (J.Q.); shejuan.an@northwestern.edu (S.A.); 3Department of Neurosurgery, Indiana University School of Medicine, Indianapolis, IN 46202, USA; nburket@iu.edu (N.B.); sccooper@iu.edu (S.C.); 4Department of Pediatrics, Children’s National Hospital, Washington, DC 20010, USA; jnazarian@childrensnational.org; 5Department of Pediatrics, Zurich Children’s Hospital, 8032 Zurich, Switzerland; 6Department of Neurosurgery, Lutheran General Hospital, Park Ridge, IL 60068, USA

**Keywords:** diffuse midline glioma, LIN28B, let-7, developmental axis

## Abstract

**Simple Summary:**

Diffuse midline glioma (DMG) is a devastating pediatric brain tumor with urgent unmet need for novel treatment modalities. LIN28B RNA binding protein is overexpressed in DMG and suppresses the let-7 family of microRNAs, which in turn suppress a plethora of oncogenes. In the present review, we summarize this LIN28B–let-7–oncogene axis across glioma subtypes and advise future research specific to DMG, offering it as a potential therapeutic vulnerability.

**Abstract:**

Diffuse midline glioma (DMG) is the most lethal of all childhood cancers. DMGs are driven by histone-tail-mutation-mediated epigenetic dysregulation and partner mutations in genes controlling proliferation and migration. One result of this epigenetic and genetic landscape is the overexpression of LIN28B RNA binding protein. In other systems, LIN28B has been shown to prevent let-7 microRNA biogenesis; however, let-7, when available, faithfully suppresses tumorigenic pathways and induces cellular maturation by preventing the translation of numerous oncogenes. Here, we review the current literature on LIN28A/B and the let-7 family and describe their role in gliomagenesis. Future research is then recommended, with a focus on the mechanisms of LIN28B overexpression and localization in DMG.

## 1. Introduction

Diffuse midline glioma (DMG) is a devastating childhood brain tumor with a median age at diagnosis of 6–7 years [1]. Between 200–300 cases are reported annually in the United States, with a two-year survival rate of less than 10 percent and a median survival of 10 months [2,3,4].

Approximately 80% of DMGs have a mutation in *HIST1H3B* or *H3F3A*, which convert position 27 on histone H3.1- or H3.3- from lysine to methionine, respectively (H3K27M mutations) [5,6,7,8]. H3 wild-type (H3WT) DMGs have enhancer of zeste homologs inhibitory protein (EZHIP) overexpression [9]. Both the H3K27M mutations and EZHIP overexpression decrease H3K27 trimethylation (H3K27me3), resulting in dysregulated gene expression, including the upregulation of endogenous retroviruses and LIN28B, an RNA binding protein (RBP) [10,11,12,13,14,15,16,17,18]. LIN28B and its paralog, LIN28A, interact with the let-7 family of microRNAs (miRs)—which includes let-7a/b/c/d/e/f/g/i/miR-98—during development and oncogenesis [19,20,21].

The biogenesis of miRs occur in the following three steps (Figure 1) [22]. First, a gene for a given let-7 family member is transcribed into a primary-let-7 (pri-let-7) transcript; this occurs in the nucleolus, a dense region of the nucleus [20]. Second, pri-let-7 is translocated to the larger, less dense region of the nucleus and processed by the microprocessor complex—which contains two proteins, DROSHA and DGCR8—into a preliminary-let-7 (pre-let-7) form [20]. Third, pre-let-7 is translocated to the cytoplasm and processed by another protein, DICER, into its mature and bioactive form, known simply as let-7 [20].

The biogenesis of the let-7 miR family is suppressed by LIN28A/B. LIN28A works with Musashi1 (MSI1) to sequester pri-let-7 in the nucleus [23]. LIN28A also binds pre-let-7 in the cytoplasm to block processing by DICER, as well as recruits a protein called terminal uridylyltransferase 4/7 (TUT4/7), which adds uridyl groups to the 3′ end of pre-let-7, an action that tags it for degradation [24,25,26,27,28]. In contrast, LIN28B sequesters pri-let-7 transcripts in the nucleolus, preventing them from continuing on in the biogenesis pathway [29].

After their biogenesis, mature miRs direct translational repression of their target gene mRNAs. This is mediated by the RNA-induced silencing complex (RISC). MiRs are first incorporated into RISC, then this miR-loaded RISC uses the miR nucleic acid sequence as a guide, with the functional unit binding to the 3′ untranslated region (UTR) of mRNAs through base pair complementarity [30]. This binding prevents the mRNAs from being translated by the ribosome.

Each of the let-7 family members decrease expression of genes that promote stemness, proliferation, and migration, thereby promoting differentiation programming, while LIN28A/B derepress these genes in a let-7-dependent manner to preserve a pluripotent phenotype [19,20,21]. Both LIN28A/B have also been shown to directly bind certain RNA transcripts, increasing their stability and according function or expression [19,20,21,31,32]. This provides a let-7-independent mechanism of oncogene upregulation. Initially discovered in *Caenorhabditis elegans*, LIN28A/B and the let-7 family are responsible for the timing of differentiation, with aberrancies in the elegant relative level of each species causing heterochronic developmental defects [33,34,35,36]. Indeed, high LIN28A/B levels and low let-7 levels is appropriate in early developmental contexts, but aberrantly present in many cancers, while the reverse proportion can cause neurodegeneration [37,38]. Interestingly, progressive LIN28A/B downregulation and let-7 family upregulation has similarly been reported in differentiation of oligodendrocyte precursor cells, the cell of origin in DMG, suggesting a conceptual model wherein DMG formation is caused by an arrest of this progression, potentially resulting from the H3K27M mutant epigenetic landscape (Figure 2, [39,40,41,42,43,44]).

Of note, dysregulated expression and function of LIN28A/B and the let-7 family has been characterized in other pediatric non-glioma brain tumors, including embryonal tumor with multilayered rosettes, atypical teratoid rhabdoid tumor, and medulloblastoma [45,46,47,48,49,50,51,52]. For matters of space, however, we focus exclusively on LIN28A/B and the let-7 family in gliomagenesis, specifically DMG.

Our review manuscript makes three novel contributions to the field of DMG. First, it introduces this LIN28B–let-7–oncogene axis, a promising therapeutic vulnerability. Second, it extensively reviews let-dependent and -independent mechanisms of LIN28A/B function across glioma subtypes. Third, it directs future work specific to DMG.

## 2. LIN28A/B and Let-7 Expression Correlate with Clinical Outcomes in Glioma

One consistent finding in the literature pertained to the expression pattern of LIN28A/B and let-7 family members. In glioma, LIN28A expression is increased, correlating directly with tumor grade and poor survival (Table 1, [32,53,54]). LIN28A expression is also increased in glioma cell lines [32]. Our group and others have shown that LIN28B is expressed in cell lines of diffuse intrinsic pontine glioma (DIPG), the pontine subtype of DMGs (unpublished data, [15,17]). Indeed, RNA-sequencing data shows that LIN28B is overexpressed in H3K27M mutant DMG when compared to H3WT DMG [16].

Of interest is a unique paper by Guo et al. [74] which characterized an LIN28B tumor-specific transcript (LIN28B-TST). The authors showed that LIN28B-TST is encoded upstream of the LIN28B-WT transcript in both low-grade glioma (LGG) and glioblastoma (GBM), providing several more codons to result in additional N-terminal amino acids in the LIN28B-TST protein [74]. An LIN28B-TST was not reported in DMG [74]. A c-Myc-regulated transcription initiation site controls the LIN28B-WT transcript [74]. *N-MYC* gain-of-function mutation as well as *C-MYC* overexpression are observed in DMG, and *C-MYC* downregulation may be seen in response to treatment [75,76]. Such data suggest LIN28B may mediate a c-Myc-driven subset of DMG, as has been characterized in a panel of c-Myc-driven cancer cell lines and in n-Myc-driven neuroblastoma [77,78]. About 20 kilobases (kb) upstream of the LIN28B-WT locus, a nuclear transcription factor Y subunit alpha (NFYA)-regulated alternative transcription initiation (ATI) site controls LIN28B-TST expression [74]. Similar to *C-MYC*, *NFYA* is overexpressed in DMG and may drive expression of such an LIN28B-TST [79]. Functionally, additional N-terminal amino acids of the LIN28B-TST protein increase its stability, leading to greater protein product [74]. LIN28B-TST enhances tumorigenicity in an in vivo liver hepatocellular carcinoma (HCC) cell murine model and HCC patients expressing LIN28B-TST have poorer outcomes [74]. In sum, NYFA and c-Myc regulate the transcription of alternative LIN28B transcripts in glioma (Figure 3).

Fitting with this paradigm of LIN28A/B overexpression in glioma, let-7a/b/d/f/g/miR-98 are decreased and correlate inversely with tumor grade (Table 1, [55,56,57,62,68,70,71,72,73]). Correspondingly, let-7a/b/e/f/g and miR-98 are increased in long-term-surviving patients, and let-7f correlates directly with survival [58,68]. Likewise, let-7a/b/d/f/i/miR-98 are decreased in glioma cell lines [57,63,68,71,72,73]. Moreover, in a glioma murine model, let-7a/c/d/e/g/miR-98 are decreased [56].

In a DIPG murine model driven by the expression of H3.3K27M mutant histones, LIN28B expression is increased, as is the expression of insulin-like growth factor mRNA binding protein 2 (IGF2BP2, also known as IMP2) [13]. Interestingly, however, a study investigating glioma stem cells (GSCs) did not detect either LIN28A/B and, in turn, found that let-7 family members were highly expressed [80]. Yet, let-7 targets were still upregulated, suggesting an LIN28A/B-independent mechanism was suppressing let-7 activity [80]. The authors showed IMP2 interacts with Argonaute protein 2 (AGO2), the catalytic component of RISC, at let-7 recognition elements on target mRNAs [80]. This prevented let-7-mediated mRNA translational repression to preserve the GSC phenotype, showing that IMP2 activity may compensate for low LIN28A/B expression (Figure 3, [80]).

## 3. LIN28A/B-Dependent Mechanisms Regulating Gliomagenesis

### 3.1. LIN28A Drives Aerobic Glycolysis

LIN28A is an established regulator of cancer metabolism: LIN28A binds to the mRNAs of several glycolytic and insulin signaling genes to increase their translation, thereby increasing glycolysis and glucose uptake, as well as derepresses a let-7 target that inhibits pyruvate entry into the tricarboxylic acid cycle, thereby reenforcing aerobic glycolysis [81]. Fitting with these data, one study of glioma cells showed that LIN28A stabilized the pre-long noncoding RNA (lncRNA) small nucleolar host gene 14 (*SNHG14*) transcript [32]. Then, *SNHG14* recruits staufen double-stranded RNA binding protein 1 (STAU1) to the interferon regulatory factor 6 (*IRF6*) mRNA 3′ UTR, in turn recruiting UPF1, an RNA helicase [32]. UPF1 mediates *IRF6* mRNA degradation, decreasing *IRF6* expression [32]. This derepresses IRF6-silenced genes, pyruvate kinase M2 (*PKM2*) and glucose transporter 1 (*GLUT1*) [32]. PKM2 and GLUT1 increase aerobic glycolysis, which in turn increases tumor microenvironment (TME) acidification [32]. Consistent with this paradigm, the authors show that decreasing LIN28A or *SNHG14*, or increasing IRF6, decreases aerobic glycolysis and proliferation [32]. They validate these results using murine subcutaneous and orthotopic xenograft models [32]. Altogether, these data suggest LIN28A changes the metabolic profile of glioma cells (Figure 4).

### 3.2. LIN28A Inhibits Apoptosis and Drives Proliferation and Migration

Because it derepresses a plethora of oncogenes through let-7 suppression, LIN28A induces a stem-like phenotype characterized by resistance of apoptosis and increased proliferation and migration. Downregulation of LIN28A also increases apoptosis and G1 phase cell count while decreasing S phase cell count and colony formation of glioma cells [54]. This may be related to the findings of one study showing that LIN28A expression correlates directly with stem cell factor expression [53]. These factors were octamer-binding transcription factor (TF) 4 (*OCT4*), high-mobility AT-hook 2 (*HMGA2*), and snail family transcriptional repressor 1 (*SNAI1*) [53]. Another important takeaway from this work was that LIN28A inhibition decreases invasion and growth of glioma cells [53]. Moreover, transduction of LIN28A into these cells decreases let-7b and let-7g expression [53]. Correspondingly, LIN28A transduction into human neural stem cells carrying dominant negative TP53, constitutively active K-RAS, and constitutively active human telomerase reverse transcriptase showed increased growth [53]. The authors validate these results using an orthotopic xenograft intracranial murine model, noting intraparenchymal invasion and a dependency on LIN28A [53]. Taken together with Section 3.1, there are important considerations to LIN28A as a factor sitting at the intersection of a multitude of oncogenic processes.

### 3.3. LIN28B Inhibits Apoptosis and Drives Proliferation

LIN28B mechanistic studies in DMG are lacking. However, some preliminary data are suggesting that it may resist apoptosis while also driving proliferation. Short hairpin RNA-mediated LIN28B knockdown has been shown to decrease proliferation and increase apoptosis of DIPG cells [15]. An additional study showed that LIN28B knockdown decreases proliferation of human embryonic stem cell-derived neural progenitor cells generated by constitutive activation of PDGFRɑ, *TP53* knockdown, and H3.3K27M mutation [82]. Therefore, it is likely that similar to LIN28A, LIN28B increases proliferation and cell viability in glioma.

## 4. Mechanisms of Glioma Cell Suppression by Let-7

### 4.1. Let-7 Drives Apoptosis and Inhibits Proliferation and Migration

In line with the conceptual model proposed (Figure 2), the let-7 family performs opposite functions in glioma cells than LIN28A/B, and this is mediated by the translational repression of a plethora of oncogenes (Table 1). Given the intersecting data below, it appears clear that we cannot tease apart these functions independently of each other, likely due to the many targets of let-7 family members.

To begin, neurotensin (NTS) is a peptide neurotransmitter and neuromodulator of the CNS that has been reported as a mediator of diverse pathologies, including Parkinson’s disease, schizophrenia, and various cancers [83]. In glioma, expression of the high-affinity NTS G-protein-coupled receptor 1 (NTSR1) correlates directly with tumor grade and indirectly with survival [84]. Mechanistically, NTSR1 increases *C-MYC* expression, extracellular signal-regulated kinase 1/2 (ERK1/2) phosphorylation, as well as the proliferation and migration of glioma cells [84,85]. In turn, treatment with the NTSR1 inhibitor, SR48692, decreases growth of a murine glioma model [84,85]. Interestingly, glioma cells treated with SR48692 have increased let-7a-3 expression, as well as activity of the pro-apoptotic protein, caspase 3 [59]. Let-7a-3 binds *Bcl-w* mRNA 3′ UTR to decrease expression of this anti-apoptotic protein, thereby inducing apoptosis [59,60]. Moreover, SR48692-mediated NTSR1 inhibition decreases *C-MYC* expression, which decreases *LIN28A* expression [59]. These results were validated in a murine glioma model [59].

In another study, ultraviolet light-induced apoptosis of glioma cell lines is increased by miR-98 treatment through a mechanism dependent on inhibitor of nuclear kappa-B kinase epsilon (IKBKE, also known as IKKi or IKKε), nuclear factor kappa-light-chain-enhancer of activated B cells (NF-κB), and Bcl-2 (Figure 5, [72]). IKBKE is a protein that mediates several proliferative and migratory signaling pathways in various cancers [86]. IKBKE overexpression has been reported in glioma and NF-κB, a transcription factor well-established to increase a stem cell-phenotype, migration, radiation resistance, aerobic glycolysis, and angiogenesis in GBM [87,88]. NF-κB increases expression of the anti-apoptotic protein, Bcl-2 [87,89]. Of note, miR-98 binds *IKBKE* mRNA 3′ UTR to decrease *IKBKE* expression, which decreases *NF-κB p50* subunit expression [72]. This leads to a decrease in NF-κB activity, causing a decrease in *Bcl-2* expression, ultimately resulting in an increase in caspase 3 activity [72]. Similarly, miR-98 decreases migration of glioma cells by binding to *IKBKE* mRNA 3′ UTR to decrease *IKBKE* expression, decreasing NF-κB p65 subunit nuclear translocation, which decreases expression of *matrix metalloproteinase* (*MMP*)*-9* [71]. *MMP-9* expression correlates directly with tumor grade and indirectly with survival in glioma [90]. Mechanistically, MMPs are zinc-dependent endopeptidases that cleave extracellular matrix (ECM) components, thereby remodeling the TME to permit invasion [91]. MMP-9 also functions through a non-canonical mechanism to increase proliferation in glioma [90]. Finally, let-7b/i decreases migration and proliferation of glioma cells by binding to *IKBKE* mRNA 3′ UTR to decrease *IKBKE* expression, which increases *E-cadherin* (*E-cad*) expression [63]. E-cad is a cell–cell adhesion protein that restricts tumor cell migration and is commonly downregulated in glioma [92]. The let-7 family, therefore, acts through *IKBKE* to alter apoptotic and migratory signals (Figure 5).

The RAS family is also a target of let-7. RAS family members are membrane-bound guanosine triphosphate (GTP)-activated binary molecular switches that initiate signaling pathways in diverse pathologies, including psychiatric and developmental conditions, as well as various cancers [93]. In DMG, the H3K27M mutation increases RAS activity, driving the RAS pathway component ERK5 to stabilize c-Myc [94]. Treatment with let-7g decreases proliferation and migration of glioma cells by binding to *pan-RAS* mRNA 3′ UTR, *N-RAS* mRNA 3′ UTR, and *K-RAS* mRNA 3′ UTR to decrease *pan-RAS*, *N-RAS*, and *K-RAS* expression, while treatment with let-7a achieves similar results by targeting *K-RAS* [55,69]. Each of these results were validated in a murine glioma model [55,69]. Therefore, *RAS* is another target of let-7 in glioma and mediates decreases in proliferation and migration.

Emerging data are shining light on *H19* lncRNA in glioma, which, similarly to LIN28A/B, works by suppressing let-7 to derepress let-7 targets. A recent study showed that in DIPG cells, *H19* lncRNA expression is increased [61]. Meanwhile, let-7a-5 treatment decreases proliferation [61]. However, *H19* decreases let-7a-5 levels via lncRNA:miR complementarity, which increases expression of let-7a-5 targets (Figure 6, [61]). It was shown that *sulfatase 2* (*SULF2*) and *oncostatin M receptor* (*OSMR*) expression is increased [61]. *SULF2* is expressed in GBM and drives PDGFRɑ activation [95]. *OSMR* is expressed in GBM, owing in part to hypoxia-induced annexin A2 (ANXA2) activation of STAT3, which drives its expression [96]. Of note, hypoxia is a feature of DMG [97]. OSMR then mediates signaling of the macrophage-associated cytokine oncostatin M to activate signal transducer and activator of transcription 3 (STAT3), leading to the upregulation of genes that increase migration, proliferation, and angiogenesis [96,98,99,100]. OSMR is translocalized to the mitochondrial matrix and interacts with the electron transport chain component NADH ubiquinone oxidoreductase 1/2 (NDUFS1/2) to upregulate oxidative phosphorylation and confer radiation resistance [101]. The same mechanism of *H19*-mediated let-7 antagonism has also been shown in human embryonic kidney 293 cells [102]. Therefore, more work is warranted to further characterize *H19* in glioma and how it blunts let-7-mediated proliferative suppression.

Let-7 also acts through cell-cycle regulators to control proliferative signaling. Treatment with let-7b decreases proliferation of glioma cells by binding to *E2F2* mRNA 3′ UTR to decrease *E2F2* expression [64]. *E2F2* expression correlates directly with tumor grade and indirectly with survival in glioma [103]. E2F2 is a TF that promotes cell-cycle progression [104]. Let-7b decreases cell migration while increasing apoptosis and S phase cell count by binding to *cyclin A2* (*CCNA2*) mRNA 3′ UTR, *cyclin B2* (*CCNB2*) mRNA 3′ UTR, polo like kinase 1 (*PLK1*) mRNA 3′ UTR, and aurora A kinase (*AURKA*) mRNA 3′ UTR to decrease *CCNA2*, *CCNB2*, *PLK1*, and *AURKA* expression [65]. Bioinformatics analyses have revealed that these are “hub” genes which coordinate cell-cycle pathways and can likely serve as biomarkers for GBM, bringing even greater importance to understanding how let-7 interacts with them [65,105,106,107].

*HMGA2* is a major target of the LIN28A/B–let-7–oncogene axis, as it changes migratory programming by altering transcriptional dynamics. HMGA2 is a non-histone chromatin-associated protein that binds to the minor groove of DNA, bending it and, thereby, regulating its accessibility during a myriad of processes, most notably transcription [108]. *HMGA2* expression correlates directly with tumor grade and indirectly with survival in glioma [109]. Raf-1 kinase inhibitor protein (RKIP) decreases invasion of glioma cells by increasing miR-98 expression, which binds *HMGA2* mRNA 3′ UTR to decrease *HMGA2* expression [70]. HMGA2 increases glioma cell migration in part by increasing *MMP-2* [110,111]. Indeed, one study assessed glioma cell colonies’ migratory rim cells to migration-restricted core cells, showing that let-7a/b/c/d/e/f/g/i are decreased in the migratory population [112]. Together, these data show that *HMGA2* is a vital target of let-7 in glioma, especially when taken with the LIN28A data discussed in Section 3.2.

Let-7 also targets pre-B-cell leukemia homeobox 3 (*PBX3*) in glioma, a protein which mediates transforming growth factor β (TGF-β) signaling, driving the expression of genes that increase migration—N-cadherin (*N-cad*), zinc finger E-box-binding homeobox 1 (*ZEB1*), *SNAI2*, and *CD44* [66]. PBX3 activates the MEK/ERK1/2 pathway, which increases c-Myc-mediated LIN28A expression [66]. As such, a positive feedback loop reinforces the migration [66]. Both let-7b and miR-98 decrease migration of glioma cells by binding to *PBX3* mRNA 3′ UTR to decrease *PBX3* expression [66,73]. All results were validated in murine glioma models [66,73]. These data offer that PBX3 is another important factor to consider in the anti-migratory signaling of let-7.

Another study found that STAT3 mediated the apoptotic effects of a different subset of let-7 family members. STAT3 is the TF component of the canonical Janus kinase (JAK)/STAT signaling pathway, directing expression of genes involved in anti-apoptosis, migration, angiogenesis, and immune suppression in GBM [113]. However, in a *PTEN*-deficient genetic background—such as that reported in a subset of DMGs—STAT3 may actually be tumor suppressive [114]. Such conflicting data warrant more work to elucidate the context-specific roles of STAT3. Regardless, treatment with let-7a-1/d/f-1 decreases proliferation while increasing apoptosis and autophagy of glioma cells by binding to *STAT3* mRNA 3′ UTR to decrease *STAT3* expression [57]. In turn, this also causes a decrease in *Bcl-2* expression, leading to an increase in c-caspase-3 activity [57]. STAT3 is, therefore, an emerging target of let-7 in glioma and, fitting with its broad biological functions, may be in part responsible for the broad glioma suppressive functions of let-7.

An additional mediator of let-7-driven caspase 3 upregulation is cyclin D1 (*CCND1*), which phosphorylates and thereby inactivates retinoblastoma (Rb) protein, a tumor suppressor that inhibits G_1_-S phase progression [115]. *CCND1* expression correlates directly with tumor grade, increases proliferation, and contributes to temozolomide resistance [116]. Treatment with let-7b rescues cisplatin sensitivity of cisplatin-resistant glioma cells by binding to *CCND1* mRNA 3′ UTR to decrease *CCND1* expression, which increases caspase 3 activity, apoptosis, and G_1_ phase cell count [67]. A partner of cyclin D1 is cyclin E (encoded by *CCNE1*), which is also responsible for Rb phosphorylation, and thus, cell-cycle stimulation [117]. Treatment with let-7f decreases proliferation while increasing G_1_ phase cell count and apoptosis of glioma cells by decreasing *CCND1*, *CCNE1*, and *Bcl-2* expression, increasing *P21*, *P27*, and *Bax* expression, and increasing caspase-3 activity [68]. p21 and p27 block cell-cycle progression by inhibiting the cyclin E:cyclin-dependent kinase 2 complex, while Bax promotes apoptosis by piercing the mitochondrial membrane to release the caspase-3 activator cytochrome-c [118,119,120]. Let-7f also decreases migration and invasion by decreasing *MMP-2* and *MMP-9* expression [68]. Similar to *MMP-9*, *MMP-2* expression correlates directly with tumor grade and indirectly with survival in glioma [121]. MMP-2 increases invasion by ECM degradation and increases growth by stimulating angiogenesis [122]. Finally, let-7f decreases proliferation, migration, and invasion by binding to *periostin* mRNA 3′ UTR to decrease *periostin* expression [68]. Much work has been put into elucidating the roles of periostin, a non-structural ECM protein, in various cancers [123]. As with many oncogenes targeted by the let-7 family, periostin expression correlates directly with tumor grade and indirectly with survival in glioma [124,125]. Mechanistically, periostin recruits M2 tumor-associated macrophages to increase growth, as well as migration, proliferation, and angiogenesis [124,125,126,127,128,129]. The aforementioned targets of let-7f were validated using a murine glioma model [68]. Altogether, let-7 controls apoptotic, proliferative, and migratory signaling in glioma.

### 4.2. Let-7 Drives Tumor-Suppressive Paracrine Signaling

Mounting evidence suggests that let-7 family members are packaged and exported out of the glioma cell, then function as ligands in tumor-suppressive paracrine signaling. Let-7a/b/e/f/g/miR-98 contain exosome-packaging motifs, and let-7b is enriched in microvesicles isolated from glioma cell media relative to its intracellular level (Figure 6, [58,130]). One study looked at exosomes released by glioma-associated stem cells (GASCs) isolated from LGGs that had undergone anaplastic transformation before four years (labeled aggressive) or after seven years (labeled less aggressive). Let-7a/e/f were downregulated in both sets of GASCs [131]. Let-7d/g/miR-98 were downregulated exclusively in the less aggressive set [131]. Let-7a-3 (a subtype of let-7a) was also downregulated exclusively in the aggressive set [131]. Interestingly, it has also been shown that pre-let-7a-3 and its murine orthologue, pre-let-7c-2, are the only let-7 family members to escape LIN28A/B regulation [132]. Pre-let-7c-2 contains a CUCUG sequence at its short apical stem loop/preE loop junction, which impairs LIN28A/B cold shock domain binding at the preE [132]. This sequence is expected to be conserved in pre-let-7a-3 [132]. This prompts intrigue into future let-7a-3 research.

Another study found that let-7 family members function in a UUGU motif-dependent manner, as toll-like receptor 7 (TLR7) ligands to regulate tumor microglia: treatment with let-7b/e decreases growth of a murine glioma model by increasing microglial infiltration in a TLR7-dependent manner [56]. This same study showed that let-7b increases caspase-3 activation and apoptosis [56]. Let-7b/c/e/f/g increase the release of pro-inflammatory signaling molecules (cytokines), namely TNF-α, IL-6, IL-10, IL-1b, GRO-a, MIP-2, and RANTES [56]. Let-7b/e increase antigen presentation mediated by CD54 and MHC1 [56]. Lastly, let-7b/d/e treatment increases migration of the microglia into the tumor bed, commonly referred to as microglial infiltration [56]. Such preliminary evidence of let-7 as a paracrine factor offers a new and exciting direction for let-7 research in glioma.

## 5. Future Directions

Across glioma subtypes, LIN28A/B are overexpressed, let-7 family members are underexpressed, and let-7 family members antagonize a plethora of oncogene mRNAs to mediate glioma cell suppression (Figure 2). These steams of evidence should motivate further investigation of the LIN28B–let-7–oncogene axis in DMG.

The mechanisms driving LIN28B overexpression must be elucidated. On the genomic level, the presence of an LIN28B-TST has not been investigated in DMG [74]. If such an LIN28B-TST exists, its mRNA transcript may share the same increased stability, and accordingly, high expression, as in hepatic adenocarcinoma [74]. On the epigenomic level, there are three areas of interest: DNA methylation and histone modifications at the *LIN28B* locus, as well as regulation of the *LIN28B* mRNA transcript. Experimentally lowered DNA methylation of the AluJb promoter increases AluJb-LIN28B fusion protein expression and higher DNA methylation reduces its expression in leukemia cells [133]. Similarly, lower DNA methylation at four CpG sites has been observed at the *LIN28B* promoter in gastric cancer, leading to higher *LIN28B* expression and increased proliferation and migration [134]. A lack of H3K27me3 and surplus of H3K27ac at the LIN28B locus in H3K27M mutant DIPG cell lines has been reported [135]. The histone deacetlylase (HDAC) sirtuin 6 (SIRT6) has been shown to act on the *LIN28B* locus to remove K3K9ac/56ac motifs, decreasing *LIN28B* expression and suppressing pancreatic ductal adenocarcinoma [136]. Of note, *SIRT6* is downregulated in GBM cell lines [137]. Together, these data suggest that canonical DNA methylation and histone post-translational modifications contribute to transcriptional regulation of *LIN28B* expression. Furthermore, the *LIN28B* mRNA transcript can be stabilized by LIN28B binding, resulting in its enhanced translation [31]. Meanwhile, miR-203 has been shown to repress its translation through *LIN28B* mRNA 3′ UTR binding in non-small cell lung cancer [138]. Unsurprisingly, miR-203 expression is inversely correlated with tumor grade in glioma [139]. Its downregulation is associated with imatinib-resistance and induction of the epithelial-mesenchymal transition (EMT) in GBM—effects mediated by the corresponding upregulation of another of its targets, SNAI2 [140]. Accordingly, miR-203 is an attractive therapeutic candidate for repressing *LIN28B* mRNA, and warrants further investigation.

The mechanisms driving LIN28B localization must also be elucidated. LIN28B principally localizes to the nucleolus, its cellular compartment of canonical let-7 antagonism [29]. It has been shown in pancreatic cancer that K-RAS can drive protein kinase C β (PKCβ) to phosphorylate LIN28B at serine 243 (S243), promoting translocation, which leads to decreased let-7i expression and increased expression of the let-7i target, *TET3* (Figure 7, [141]). TET3 then catalyzes the conversion of 5-methylcytosine (5mC) to 5-hydroxymethylcytosine (5hmC), resulting in global DNA demethylation which subsequently mediates an increase in *LIN28B* expression, facilitating a positive feedback loop [141]. Indeed, *K-RAS* amplification, *TET3* overexpression, low 5mC, and high 5hmC have all already been reported in DIPG [75,142,143,144]. PKCβ has been discussed and targeted as a driver of angiogenesis, proliferation, and survival in GBM [145]. Neurofibromatosis 2 (*NF2*) also regulates LIN28B translocation (Figure 7, [146]). High cell density (cell contact) drives NF2 dephosphorylation at S518, enhancing its association with LIN28B [146]. This sequesters LIN28B in the cytoplasm, leading to higher let-7a/c/g expression [146]. NF2 is glioma-suppressive, increasing large-tumor suppressor signaling and decreasing canonical and non-canonical Wnt signaling [147]. Similarly, NF2 decreases glial cell proliferation by decreasing ErbB2-dependent Src-FAK-paxillin signaling [148]. The *NF2* locus is hypermethylated and underexpressed in GBM [147,149]. Altogether, these data suggest that restoration of NF2 expression is a potential strategy for DMG treatment. However, a high level of S518-phosphorylated NF2 correlates with high *NOTCH1* and *EGFR* expression in GBM, promoting proliferation [150]. Thus, both the quantity and relative phosphorylation status of NF2 require consideration. In sum, therapeutic potential lies in decreasing the translocation of LIN28B from the cytoplasm to the nucleolus.

Finally, we must consider the potential for let-7-independent mechanisms of LIN28B expression and its effects in DMG. To the authors’ knowledge, there are currently no publications on let-7-independent mechanisms of LIN28B expression effects in glioma. However, LIN28B has been shown in other cancers to bind the 3′ UTR of oncogene mRNA transcripts, increasing their stability and, accordingly, their translation. In this way, LIN28B promotes transformation and migration in colon cancer through *LGR5*, *PROM1*, and *CDX2* mRNAs, promotes stemness and EMT in gastric cancer through *NRP-1* mRNA, inhibits apoptosis in ovarian cancer through *AKT2* mRNA, and promotes migration in neuroblastoma through MYCN-induced mRNAs [151,152,153,154,155]. In addition, LIN28B promotes migration and proliferation in cholangiocarcinoma through TGF-β-induced protein (TGFBI), although a mechanistic relationship between LIN28B and *TGFBI* mRNA 3′ UTR has not yet been verified [156]. Altogether, searching for let-7-independent mechanisms of LIN28B expression in DMG could be promising, potentially offering additional targets in a wider network of LIN28B-driven gliomagenesis.

## 6. Conclusions

Here, we provide a detailed review of the expression and function of the LIN28A/B–let-7–oncogene axis in gliomagenesis, including DMG formation. Given these data, additional efforts to elucidate LIN28B expression and localization mechanisms, as well as its functional utility as a therapeutic target, in DMG should be considered.

## Figures and Tables

**Figure 1 cancers-15-03241-f001:**
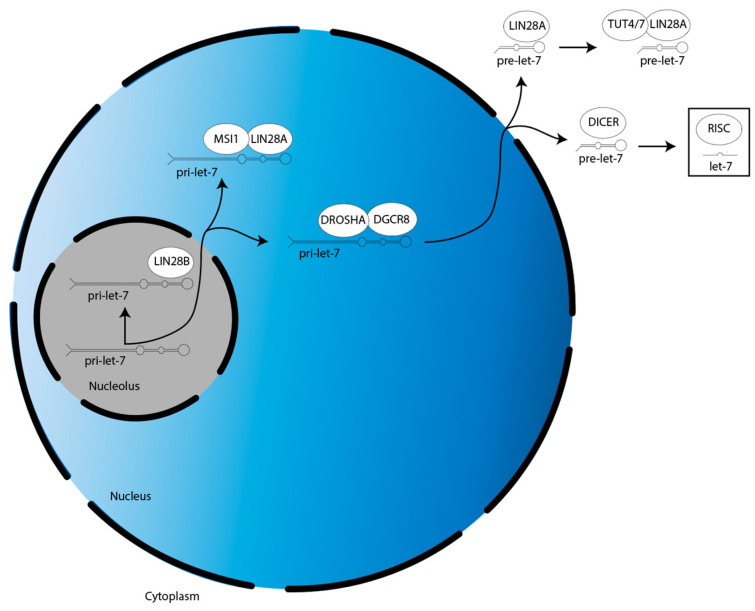
Overview of LIN28A/B activities in the suppression of let-7 miR family member biogenesis. There are two splicing events during let-7 biogenesis: pri-let-7 is spliced by the microprocessor (DROSHA:DGCR8) into pre-let-7 in the nucleus, then pre-let-7 is spliced by DICER in the cytoplasm. Let-7 is then incorporated into RISC to direct translational repression. LIN28A cooperates with MSI1 and TUT4/7 to suppress let-7 biogenesis in the nucleus and cytoplasm, respectively. LIN28B suppresses let-7 biogenesis individually in the nucleolus.

**Figure 2 cancers-15-03241-f002:**
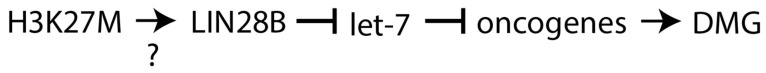
A conceptual model for the LIN28B–let-7–oncogene axis in DMG. The H3K27M epigenetic landscape may drive LIN28B expression, which decreases mature let-7 levels, which increases oncogene expression, facilitating DMG formation. This would make the LIN28B–let-7–oncogene axis a mechanistic mediator of H3K27M mutant DMG. Further investigation into the necessity and sufficiency of the H3K27M mutation to promote LIN28B expression in DMG is needed, as it may be the case that even in H3WT tumors LIN28B is upregulated, thereby making H3K27M oncohistones dispensable. DMG, diffuse midline glioma.

**Figure 3 cancers-15-03241-f003:**
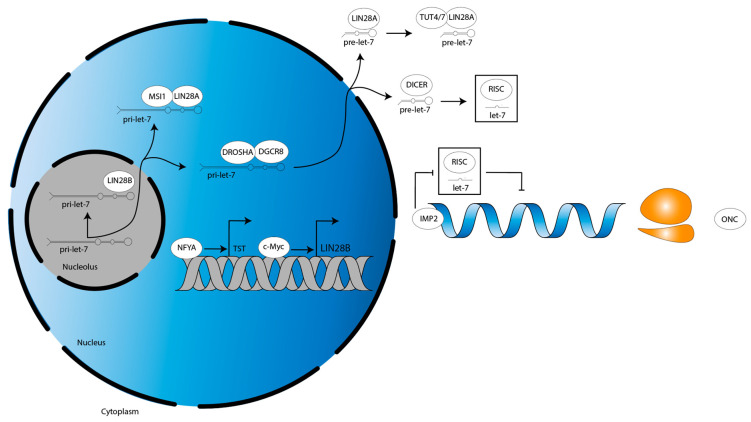
LIN28B transcriptional dynamics and let-7 functional antagonism in glioma. NFYA and c-Myc drive the transcription of LIN28B-TST and LIN28B-WT, respectively. Meanwhile, IMP2 blocks function of let-7-loaded RISC to derepress let-7 target oncogenes, thereby recapitulating the cellular phenotype induced by let-7-dependent LIN28A/B function. LIN28B-TST, LIN28B tumor-specific transcript. LIN28B-WT, LIN28B wild-type transcript. ONC, oncogene. Grey double helix, DNA. Blue single helix, mRNA. Orange two-subunit structure, ribosome.

**Figure 4 cancers-15-03241-f004:**
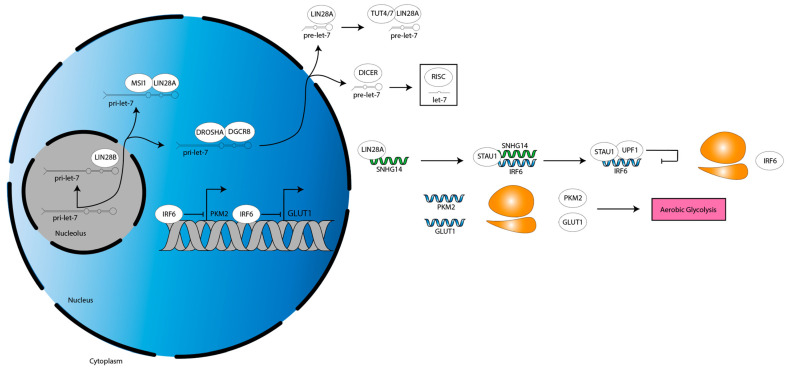
Metabolic activities downstream of LIN28A. LIN28A stabilizes *SNHG14*, driving an STAU1- and UPF1-dependent pathway that downregulates *IRF6* expression. This derepresses IRF6-silenced genes *PKM2* and *GLUT1*, resulting in an upregulation of aerobic glycolysis. Grey double helix, DNA. Blue single helix, mRNA. Green single helix, lncRNA. Orange two-subunit structure, ribosome. Pink box, programming.

**Figure 5 cancers-15-03241-f005:**
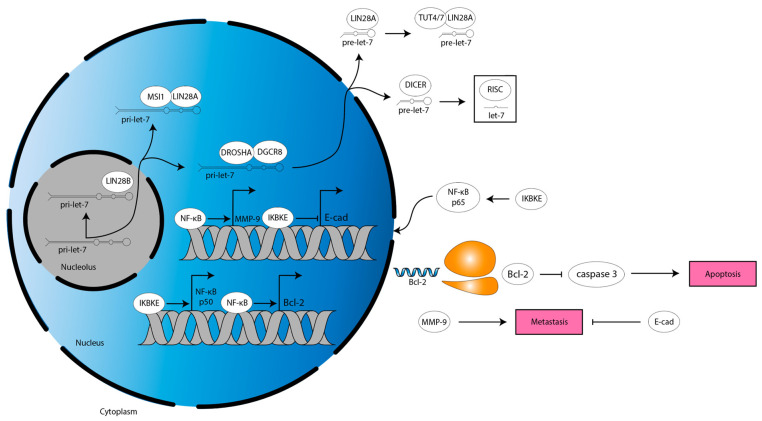
Anti-apoptotic and metastatic mechanisms downstream of let-7 target IKBKE in glioma. IKBKE drives NF-κB-mediated *MMP-9* and *Bcl-2* expression while also downregulating *E-cad* expression. This inhibits apoptosis and facilitates metastasis. Blue single helix, mRNA. Orange two-subunit structure, ribosome. Pink box, programming.

**Figure 6 cancers-15-03241-f006:**
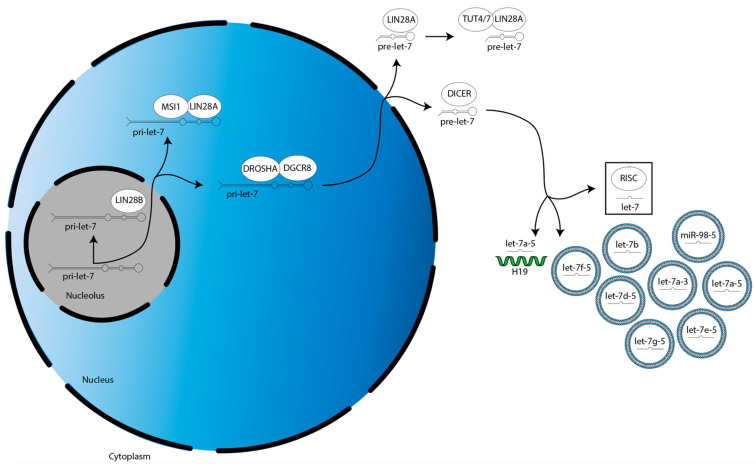
Mechanisms of mature let-7 cytoplasmic sequestration or exportation in glioma. Mature let-7a-5 is bound by *H19* while several let-7 family members are packaged for export. Each mechanism decreases the amount of let-7 loaded into RISC, thereby decreasing canonical let-7 activity. Green single helix, lncRNA.

**Figure 7 cancers-15-03241-f007:**
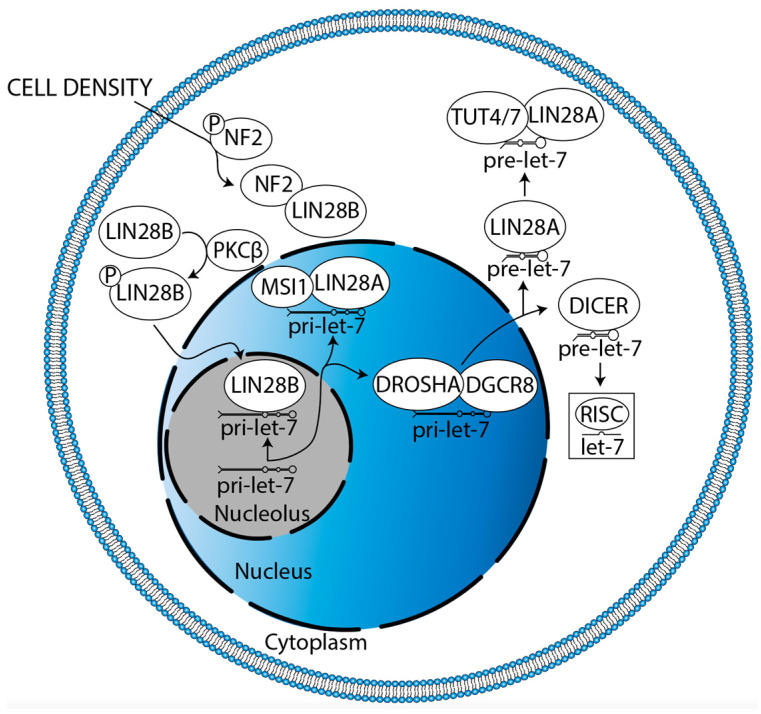
Potential mechanisms regulating LIN28B nucleolar localization in glioma and cytoplasmic sequestration in glioma. K-RAS drives PKCβ-mediated LIN28B phosphorylation, facilitating its nucleolar translocation. Meanwhile, cell density dephosphorylates NF2, enhancing its association with LIN28B to result in cytoplasmic sequestration. Each mechanism must be verified in glioma.

**Table 1 cancers-15-03241-t001:** A review of current knowledge on LIN28A/B and let-7 family members in glioma. LIN28A/B are overexpressed in glioma while let-7 family members are underexpressed and mediate the translational repression of a plethora of oncogene mRNAs. ND, no data.

Species	Expression in Glioma Samples	Expression in Glioma Models	Clinical Correlations	Targets in Glioma	References
LIN28A	Increased	Increased	Higher grade, poor survival	ND	[32,53,54]
LIN28B	Increased	Increased	ND	ND	[13,15,16,17]
let-7a	Decreased	Decreased	Lower grade, better survival	*Bcl-w*, *K-RAS, SULF2*, *OSMR*, *STAT3*	[55,56,57,58,59,60,61]
let-7b	Decreased	Decreased	Lower grade, better survival	*IKBKE*, *E2F2*, *CCNA2*, *CCNB2*, *PLK1*, *AURKA*, *PBX3*, *CCND1*	[58,62,63,64,65,66,67]
let-7c	ND	Decreased	ND	ND	[56]
let-7d	Decreased	Decreased	ND	*STAT3*	[57]
let-7e	ND	ND	Lower grade, better survival	ND	[58]
let-7f	Decreased	Decreased	Lower grade, better survival	*STAT3*, *CCND1*, *CCNE1*, *Bcl-2*, *MMP-2*, *MMP-9*, *periostin*	[57,58,68]
let-7g	Decreased	Decreased	Lower grade, better survival	*pan-RAS*, *N-RAS*, *K-RAS*	[56,58,69]
let-7i	Decreased	Decreased	Lower grade	*IKBKE*	[63]
miR-98	Decreased	Decreased	Lower grade, better survival	*IKBKE*, *HMGA2*, *PBX3*	[56,58,70,71,72,73]

## Data Availability

No new data were created or analyzed in this study. Data sharing is not applicable to this article.

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
