# Peer review of "LIN28B and Let-7 in Diffuse Midline Glioma: A Review"

_cancers, 2023, doi:10.3390/cancers15123241_

Round 1

Reviewer 1 Report

Comments and Suggestions for Authors:

This is a nice review that covers some fascinating papers suggesting that a LIN28/let-7 axis may be dysregulated in glioma and play an important role in DMG and other brain tumors. This review is likely to be of interest to the readers of Cancers - especially researchers and physicians with a focus on pediatric cancer, RBPs, and on understanding the mechanisms of brain tumor development and progression. RBPs and miRs have received far less coverage than other regulators of DMG gliomagenesis like histone modifying enzymes, so this review offers an important perspective.

However, there are a number of areas where the writing, organization, and figures could be improved. Some of the sections of this review come across more like laundry lists of studies rather than as attempts to synthesize data from multiple papers into key concepts. For each subsection of this review, it would be helpful to include an opening sentence or two that introduces the main concepts and 1-2 concluding lines to summarize the main findings of the subsection (“take home message”). Along the same lines, re-organizing each paragraph around a common theme and including a topic and concluding sentence to highlight the main points would improve the writing flow. Careful consideration should also be given to transitions between different paragraphs and on establishing a logical progression of ideas from one section to the next. Point by point critiques and suggestions are outlined, below:

Point by point critique:

In lines 74-76, the authors state that let-7 and LIN28A/B are called “maturation” factors and “pluripotency” factors because of functions in controlling “proliferation and migration.” These statements are confusing since let-7 and LIN28A/B may have roles in different aspects of cell biology (stemness, differentiation, proliferation, and migration). More generally, while the opening paragraphs do a nice job of introducing DMG and other H3K27-altered tumors and let-7 miR processing, an introduction to the biological functions of LIN28 and let-7 seems to be lacking. It would be helpful to include an additional paragraph or two in the intro providing an overview of the known functions of LIN28/let-7 in cancer, stem cells, development and so on. citing previous review articles

In lines 81-82, it is not clear what is meant by “canonical and non-canonical axis mechanisms”.

let-7a/b/d/e/f/g/miR-98 are not introduced until line 110. Defining the let-7 miR family members earlier, when first mentioned would make sense. Some of the different transcripts listed in lines 126-136 and elsewhere are difficult to visualize and understand. Can you include a table of the let-7 transcripts in humans?

The interpretation of reference 46 in lines 116-118 is confusing. This paper seems to demonstrate that (1) LIN28 expression is reduced in GSCs leading to the up-regulation of let-7 and (2) IMP2 is co-expressed with let-7 in GSCs and blocks let-7 dependent degradation of target transcripts. Is it possible to re-write this section to highlight the most important findings from this paper?

Lines 148-167 describing a connection between LIN28A and glycolysis could be simplified to highlight the most relevant points. Are there other papers that point to roles for LIN28 in metabolism (in cancer, or in glial cells) that could be cited here and described in the same paragraph?

The summaries of key papers in Sub-section 5 describing connections between let-7 and NTS signaling, IKBE, and so on are quite convoluted and could similarly be edited to highlight the most pertinent conclusions and molecular mechanisms.

Notes on organization and writing:

Including more descriptive section/sub-section headings would be helpful. The heading for section 2 might be changed from “LIN28A/B and let-7 in glioma” to something like “LIN28A/B and let-7 expression correlate with clinical outcomes in glioma.”

Similarly, section 4 might be changed from “Mechanisms of LIN28A/B in gliomagenesis” to something to the tune of, “LIN28A/B-dependent mechanisms regulating gliomagenesis”. Re-organizing subsection 4 into paragraphs covering roles for LIN28 in controlling glioma metabolism, apoptosis, and transformation/proliferation would make this section easier to follow as a reader.

Section 5. could similarly be re-organized so that each paragraph centers on a specific glioma phenotype or cell behavior and then describes current knowledge of roles for let-7 family miRs in regulating those phenotypes. From the current version of the manuscript, possible sections/paragraphs might include, let-7 functions in glioma apoptosis, proliferation, NFKappaB signaling, migration, and the regulation of the tumor microenvironment.

Be careful about absolute or vague language:

The first section on miRNA processing includes multiple statements with the phrase “all miRs”, which could be misleading. What if there are some unique miR processing pathways that have not yet been characterized?

Line 89, remove “important”.

Line 116, please change “were unexpressed” to indicate that LIN28A/B were “not detected” in that study, since it is possible that their methods were not sensitive enough.

Line 173, remove “significant”, unless you mean statistically significant

Line 311, remove “robust.”

Figures:

In the current version of Figure 1, it is unclear that MSI1 is functioning to sequester pri-let-7 in the nucleus. Is there a different way that you could illustrate this concept such as including branching paths towards either productive processing versus sequestration?

The text in Figure 2 is too small to read. Figure 2 could be simplified to highlight main pathways directly linked to let-7/LIN28 functions in glioma. Another option to consider would be to make a separate figure or panel showing the upstream regulators of let-7/LIN28 and then the downstream targets of LIN28/let-7. The legend for Figure 2 should also be expanded to summarize the pathways illustrated.

References:

For the discovery/initial characterization of LIN28 and let-7 in C. elegans and humans, please cite:

Ambros V and Horvitz HR, Science (1984)

B.J. Reinhart, et al., Nature (2000)

A.E. Pasquinelli, et al. Nature (2000).

Viswanathan, S. R., et al. Science (2008).

For the discovery of histone mutations: Please cite Wu et al., Nature Genetics (2012) and Schwartzentruber et al., Nature (2012).

For the discovery that H3K27M reduces H3K27Me2/3 please also cite Lewis et al. Science (2013)

English usage is good with minor errors and only a few stylistic issues.

Author Response

Thank you Reviewer 1 for your comments. These have been addressed point by point in the attached document.

Reviewer 2 Report

This review paper about the connection of LIN28B and let-7 in diffuse midline gliomas is an in interesting paper, which gives a nice overview about the features of this pathway in DMG.

Major comment:

Aberrations of LIN28B (and let-7) is not an exclusive feature of DMG, it is also known in other pediatric non-glioma brain tumors (ETMR, ATRT, MBL, etc.) This should be also detailed in Introduction section, makes reader clear that aberration of this protein (LIN28B) is quite often in brain tumors, and in DMG this aberrant regulation has have some special features.

This paper needs more figures as there are long description of pathways which should be easier to follow with figures.

Minor comments:

Font size in Figure 2 is so little that this figure is illegible. This figure should be fitted to the readers' vision

Line 111-112: let-7 f appears in both part of the sentence: once as "increased in long term surviving patients" and next part of the sentence  "correlates inversely with survival" - this should be clarified

Author Response

Thank you Reviewer 2 for your comments. These have been addressed point by point in the attached document.

Reviewer 3 Report

Line 45-47 - suggest addition of reference(s)

Could mechanism of LIN28A/B be incorporated into Figure 1?

Line 75 - suggest addition of reference(s)

Line 87 - Usually review articles do not refer to "unpublished data", also missing ")" at end of sentence.

Line 89-92 - suggest moving to lines before line 118-121.

Line 97 - suggest removing line "This could be of note, because"

Line 106 - suggest rewording - "LIN28B-TST has been shown to enhance tumorigenicity in an in vivo liver hepatocellular carcinoma (HCC) cell murine model and  HCC patients LIN28B-TST expressing have poorer outcomes [29].

Line 116 -  suggest changing wording - "unexpressed", change to "not expressed"

Lines 110-114 - could this be summersied graphically? Let expression in relation to tumour grade and or survival outcomes? And the let expression in glioma cells vs in vivo. It is very confusing as it currently stands and a graphic may make the messaging clearer.

Section 5.1 - could you use an graphic to explain this also? This section may need to be broken down into smaller sections (5.1.1, 5.1.2 etc...) as currently there is an overwhelming amount of information.

Figure 2 - text is too small and unreadable in this current format.

Section 5.2 - suggest move sentences in lines 284-287 to top of section to introduce RAS before detailing 281-284, then 287-288.

Section 5.3 - suggest move sentence in lines 294-298 to top of section to introduce HMGA2 prior to detailing in line 290-294.

Section 5.4 - suggest introducing PBX3 prior to referring to published experimental results.

Quality of English is fair.

There is a bit of repetition of "surprisingly", "move over", "additionally", "specifically", "further" - some of these could be avoided by rewording/re-focusing some of the sentences.

Author Response

Thank you Reviewer 3 for your comments. These have been addressed point by point in the attached document.

Round 2

Reviewer 1 Report

This review by Knowles et al. has greatly improved since the first submission. However, there are still areas where the writing is difficult to follow. Some careful editing would make this review more accessible to a general audience interested in brain tumor biology and roles for LIN28/let-7 in cancer. I would also recommend continuing to improve and simplify the figures. Specific suggestions and recommendations are outlined, below.

I would suggest making some minor changes to the figures, including:

·         the font size on the figures seems very small, I would recommend increasing the size of the labels so they are easier to read

·         Is it possible to include the x-axis and y-axis labels in Figure 2C, and 2D, rather than showing these separately in Figure 2B?

·         Can you include some explanation for the model in Figure 2D, which appears to show let-7 and LIN28 levels going up and down throughout DMG disease progression? Is there any evidence that these dynamic expression changes are occurring in DMG?

·         please add references to Table 1 to show where the different clinical correlations and gene expression data were reported in the literature

·         The Figure legends could be expanded to explain key features of the pathways illustrated to complement the descriptions in the main text.

·         In Figure 3, the link between IRF6 and PKM2/GLUT1 is not clear. Can you illustrate the idea that IRF6 acts to down-regulate these genes in the diagram?

There are also some areas where I would suggest some careful editing of the writing:

·         Line 58: please check verb-subject agreement.

·         Lines 118-119: seems to be missing a word, did you mean – “..mechanisms of LIN28A/B function across glioma subtypes.

·         Line 123, remove “steadfast”

·         Regarding the paragraph starting on Line 138, I would recommend stating whether LIN28B-TST has been detected in DMG and be cautious about comparisons with cancers arising in other tissues. This isn’t made clear until many paragraphs later in the discussion of future directions.

·         Lines 169-170 could potentially be combined into one sentence.

·         Line 173, I would recommend editing to “as is the expression of insulin…”

·         Can the sentence from Lines 207-2011 be divided into two sentences? As written, it is difficult to follow.

·         Line 214, the phrase “increasing its stability” appears redundant with the first part of the sentence.

·         The description in Lines 214-218 is a bit convoluted. Could this be simplified to something like, “Stabilized SNHG14 was then shown to down-regulate IRF6 through a STAU1 and UPF1-dependent pathway resulting in the de-repression of IRF6-silenced genes like GLUT1 and PKM2” …?

·         Line 235 – Did you mean something like - “…induces a stem-cell like phenotype characterized by resistance to apoptosis and increased proliferation and migration”?

·         The paragraph starting at line 287 could be edited to focus on the main points. This section is difficult to understand. For instance, “a mechanism dependent on three factors” (Line 288), might be changed to “a mechanism dependent on IKBKE, NFkappaB, and BCL-2.”

·         Line 309, please remove “firmly”

·         Line 318-319 seems to be missing some words. Is c-MYC downstream of a RAS/ERK5 signaling pathway, or are ERK5 activation and c-MYC stabilization happening in parallel downstream of RAS?

·         The description of H19 repressing let-7 starting at Line 325 might fit better in the previous section where Lin28-dependent repression of let-7 family miRs is described. This paragraph could also be shortened to highlight the most relevant points.

·         Line 433, please remove “confident”

·         I would be cautious about the statement that let-7-a3 may be a “critical let-7 family member” (Line 498-490), since the observation of decreased let-7-a3 expression does not guarantee that this miR is playing a functional role in the more/less aggressive tumors. What do you mean by “let-7a-3 treatment” in Lines 495-496? Is there a specific strategy that you would suggest here?

·         I would recommend using official gene symbols rather than alternative gene names (e.g. NF2 versus merlin, LATS2 etc.).

Overall the English language usage is good. There are some typos, sentence structure issues, etc. that should be addressed before this review is ready for publication.

Author Response

REVIEWER 1

This review by Knowles et al. has greatly improved since the first submission. However, there are still areas where the writing is difficult to follow. Some careful editing would make this review more accessible to a general audience interested in brain tumor biology and roles for LIN28/let-7 in cancer. I would also recommend continuing to improve and simplify the figures. Specific suggestions and recommendations are outlined, below.

I would suggest making some minor changes to the figures, including:

  •       the font size on the figures seems very small, I would recommend increasing the size of the labels so they are easier to read

      We thank the reviewer for this feedback. Upon enlarging the font the figures became less crisp and easy to read. Our thoughts are that the majority of the readership will be online and perhaps can expand the figure to facilitate ease of viewing.

  •       Is it possible to include the x-axis and y-axis labels in Figure 2C, and 2D, rather than showing these separately in Figure 2B?

Thank you for this suggestion, we agreed that Figure 2 was a bit confusing and speculative. Therefore, 2B,C,D were all removed keeping only 2A in the paper for better clarity (please see next comment, below).

  • Can you include some explanation for the model in Figure 2D, which appears to show let-7 and LIN28 levels going up and down throughout DMG disease progression? Is there any evidence that these dynamic expression changes are occurring in DMG?

This was a good idea. To achieve this feedback, 2B,C,D were all removed given that 2D was more speculative than the authors thought would be appropriate.

  • please add references to Table 1 to show where the different clinical correlations and gene expression data were reported in the literature

      Done –Please see extensively revised table 1

  • The Figure legends could be expanded to explain key features of the pathways illustrated to complement the descriptions in the main text.

      Done – please see table 1 and Figures 3-7 with an emphasis on simplification of mechanisms described in the text.

  • In Figure 3, the link between IRF6 and PKM2/GLUT1 is not clear. Can you illustrate the idea that IRF6 acts to down-regulate these genes in the diagram?

      Thank you for this point. Please see the middle-left of the figure 4 (right-bottom of nucleus), wherein IRF6 downregulates PKM2 and GLUT1 transcription (inhibition arrows)

There are also some areas where I would suggest some careful editing of the writing:

  • Line 58: please check verb-subject agreement.

      Done – please see lines 58-59

  • Lines 118-119: seems to be missing a word, did you mean – “..mechanisms of LIN28A/B function across glioma subtypes.

      Done – please see line 113

  • Line 123, remove “steadfast”

      Done – please see line 115, replaced with “consistent”

  • Regarding the paragraph starting on Line 138, I would recommend stating whether LIN28B-TST has been detected in DMG and be cautious about comparisons with cancers arising in other tissues. This isn’t made clear until many paragraphs later in the discussion of future directions.

      Done – please see line 155

  • Lines 169-170 could potentially be combined into one sentence.

      Done – please see lines 202-203

  • Line 173, I would recommend editing to “as is the expression of insulin…”

      Done – please see line 195

  • Can the sentence from Lines 207-2011 be divided into two sentences? As written, it is difficult to follow.

      Done – please see line 240

  • Line 214, the phrase “increasing its stability” appears redundant with the first part of the sentence.

      Done – please see line 216

  • The description in Lines 214-218 is a bit convoluted. Could this be simplified to something like, “Stabilized SNHG14 was then shown to down-regulate IRF6 through a STAU1 and UPF1-dependent pathway resulting in the de-repression of IRF6-silenced genes like GLUT1 and PKM2” …?

      Done - please see lines 216-220. Additionally, the associated figure legend (Figure 4) is simplified and similar to the suggested phraseology.

  • Line 235 – Did you mean something like - “…induces a stem-cell like phenotype characterized by resistance to apoptosis and increased proliferation and migration”?

      Done – please see lines 235-237

  • The paragraph starting at line 287 could be edited to focus on the main points. This section is difficult to understand. For instance, “a mechanism dependent on three factors” (Line 288), might be changed to “a mechanism dependent on IKBKE, NFkappaB, and BCL-2.”

      Done – please see lines 315-318

  • Line 309, please remove “firmly”

      Done – please see line 347

  • Line 318-319 seems to be missing some words. Is c-MYC downstream of a RAS/ERK5 signaling pathway, or are ERK5 activation and c-MYC stabilization happening in parallel downstream of RAS?

      Done – please see line 359-360

  • The description of H19 repressing let-7 starting at Line 325 might fit better in the previous section where Lin28-dependent repression of let-7 family miRs is described. This paragraph could also be shortened to highlight the most relevant points.

After careful consideration, the authors felt that this description was better suited for the in-depth analyses of the main section, and not in the introduction as the reviewer kindly suggested. Our intent was to not take away focus from the description of LIN28A/B--let-7 interactions. The section on IMP2, for example, could also be considered non-canonical let-7 signaling, but once again this in-depth analysis would distract if placed into the introduction and we felt it would belong better in the main section. We did take time to edit the paragraph as suggested. 

  • Line 433, please remove “confident”

Done – please see line 471

  • I would be cautious about the statement that let-7-a3 may be a “critical let-7 family member” (Line 498-490), since the observation of decreased let-7-a3 expression does not guarantee that this miR is playing a functional role in the more/less aggressive tumors. What do you mean by “let-7a-3 treatment” in Lines 495-496? Is there a specific strategy that you would suggest here?

Done – a deletion was made to ensure that the reader would not be led astray by the treatment-related proposition that is currently not found in the literature, please see line 487-492.

  • I would recommend using official gene symbols rather than alternative gene names (e.g. NF2 versus merlin, LATS2 etc.).

Done – please see line 550-561

Reviewer 2 Report

Minor comment: Figure 1B should be merged with 1C and 1D separately for easier undestanding

Author Response

REVIEWER 2

Minor comment: Figure 1B should be merged with 1C and 1D separately for easier understanding

Authors' Response: Figure 1 does not have parts B, C or D. We think perhaps the reviewer meant to indicate figure 2B, C and D. 2B, C, and D were all removed for clarity.

Reviewer 3 Report

N/A

Author Response

Reviewer 3 had no further comments or requested edits.